# Hybrid ZnO Electron Transport Layer by Down Conversion Complexes for Dual Improvements of Photovoltaic and Stable Performances in Polymer Solar Cells

**DOI:** 10.3390/nano10010080

**Published:** 2020-01-01

**Authors:** Fanchen Bu, Wenfei Shen, Xiaolin Zhang, Yao Wang, Laurence A. Belfiore, Jianguo Tang

**Affiliations:** 1Institute of Hybrid Materials, National Center of International Joint Research for Hybrid Materials Technology, National Base of International Sci. & Tech. Cooperation on Hybrid Materials, Qingdao University, Qingdao 266071, China; bufanchen95@163.com (F.B.); jiushiwo5698@126.com (W.S.); xiaolinzhang0307@126.com (X.Z.); wangyaoqdu@126.com (Y.W.); 2Department of Chemical and Biological Engineering, Colorado State University, Fort Collins, CO 80523, USA

**Keywords:** down-conversion materials, polymer solar cells, enhance light absorption, enhance UV stability, high photovoltaic performances

## Abstract

Polymer solar cells (PSCs) have shown excellent photovoltaic performance, however, extending the spectral response range to the ultraviolet (UV) region and enhancing the UV light stability remain two challenges to overcome in the development of PSCs. Lanthanide down-conversion materials can absorb the UV light and re-emit it at the visible region that matches well with the absorption of the active layer material PTB7-Th (poly[[2,6′-4,8-di(5-ethylhexylthienyl)benzo[1,2-b;3,3-b]dithiophene][3-fluoro-2[(2-ethylhexyl)carbony]thieno[3,4-b]thiophenediyl]]) and PBDB-T-2F, thus helping to enhance the photovoltaic performance and UV light stability of PSCs. In this research, a down-conversion material Eu(TTA)_3_phen (ETP) is introduced into the cathode transport layer (ZnO) in PSCs to manipulate its nanostructure morphology for its application in hyperfine structure of PSCs. The device based on the ZnO/ETP electron transport layer can obtain power conversion efficiencies (PCEs) of 9.22% (PTB7-Th–PC_71_BM ([6,6]-phenylC_71_-butyric acid methyl ester) device) and 13.12% (PBDB-T-2F–IT-4F device), respectively. Besides, in the research on PTB7-Th-PC_71_BM device, the stability of the device based on ZnO/ETP layer is prolonged by 70% compared with the ZnO device. The results suggest that the ZnO/ETP layer plays the role of enhanced photovoltaic performance and prolonged device stability, as well as reducing photo-loss and UV degradation for PSCs.

## 1. Introduction

Polymer solar cells (PSCs) have attracted much attention owing to their advantages in making large-area flexible solar panels using low-cost solution coating techniques [1,2,3,4,5]. Recent research works have been conducted to improve the power conversion efficiency (PCE) of PSCs by the development of novel structure donor and acceptor materials. The PCEs of single-junction polymer solar cells recently scored 16.5% following the development of new polymer materials [6]. However, it is still necessary to improve photovoltaic performances in PSCs for commercial applications. The main limiting factors for the performance of PSCs are the narrow spectral response range of the organic active layer materials and the degradation of polymers [7,8,9]. As the spectral response range of polymer solar cells is mainly in the visible light region, the solar cells cannot effectively utilize solar radiation in the ultraviolet (UV) and infrared (IR) regions [10,11]. In addition, the degradation of polymers under the action of ultraviolet rays and additives causes the photovoltaic performance of organic cells to decline, which limits the conversion efficiency of PSCs [9,12,13,14,15,16]. To solve those problems, researchers have used different fabrication methods to increase the spectral response range of PSCs; different donor materials with complementary light absorptions were applied to tandem solar cells and a ternary device with two complementary absorptions donors and one acceptor has been used [17,18,19]. However, tandem and ternary PSCs can improve the light absorption effectively, but the spectral response range in the UV and IR regions is not increased, and it is more complicated in donor material selection and device fabrication, which can greatly increase processing cost. By employing the UV-cut filter mounted on the front of glass substrates to block most UV photons, the UV stability and the lifetime of the PSCs can be effectively improved, however, the total intensity of incident sunlight dropped by 20%, resulting in a decrease in efficiency [20]. Thus, it is highly desirable to explore a feasible method to enhance the solar spectral response of PSCs.

Lanthanide down-conversion materials can be utilized in photovoltaic devices [21,22]. They are an ideal candidate material for extending the spectral response of the PSCs to the UV region. Lanthanide down-conversion materials can absorb the UV light and re-emit it at visible region thanks to the specific 4f electronic structure of lanthanide ions [10,23]. Therefore, the application of lanthanide luminescence down-conversion technology to photovoltaic devices is an effective approach to enhance the photoelectric performance. In past research, lanthanide down-conversion materials have been used to enhance the photovoltaic performance and UV light stability in inorganic perovskite solar cells [21,22,24,25,26]. Thus, it is theoretically feasible to enhance the photovoltaic performance in organic photovoltaic devices by lanthanide luminescence. Wu [27] introduced lanthanide ions into the interlayer to improve the solar spectrum response in the UV region of PSCs. However, in their work, the PECs of the devices is quite low (2.96%), and the PCE has no enhancement relative to the reference device. In addition, the lanthanide ions suffer from weak light absorption, and the amount of radiation directly absorbed by the direct excitation is very limited. However, this problem can be overcome by the “antenna effect” of organic ligands [28]. Lanthanide complexes have intense absorption bands to absorb much more UV light than the lanthanide ions, and the absorbed ultraviolet energy can effectively transfer from the organic ligands to the lanthanide ion by intramolecular energy transfer, which could enhance the photovoltaic performance and UV stability of PSCs [10,29]. Europium (Eu^3+^) is one of the most efficient down-converting ions, which has high emission intensity, 1,10-phenanthroline (phen), and 2-trifluoroacetonate (TTA) as the ligands with wide UV absorption are matched well with Eu^3+^ ions [30,31]. Typically, Eu(TTA)_3_phen (ETP) luminescent complexes have irregular structures and large size differences, which are not ideal when applied to a precise structure of PSCs. Hence, it is necessary to adjust the size and distribution of ETP complexes by changing the concentration of the complexes to optimize the photovoltaic performance of PSCs.

In this work, we constructed a simple and effective way to regulated the size of solid micelles of rare earth complexes and then applied them to polymer solar cells to increase the photoelectric conversion efficiency. We choose ZnO as the cathode buffer layer of inverted structure devices, which effectively avoids the effect of corrosive and hygroscopic hole transport poly (3,4-ethylenedioxythiophene): poly (styrene sulfonate) (PEDOT/PSS) on the stability of solar cells [32,33]. To manipulate the nanostructure morphology of Eu(TTA)_3_phen for its application in PSCs, down-conversion material Eu(TTA)_3_phen is introduced into the cathode transport layer (ZnO) by the solution method. By careful characterizations, the ETP complexes are excited by UV light and re-emit visible light that matched well with the absorption of the active layer material (PTB7-Th–PC_71_BM and PBDB-T-2F–IT-4F), thus helping to enhance photocurrent density of PSCs. In addition, the absorption of UV light by ETP complexes reduces the effect of UV light on the active layer materials and enhances the UV stability and lifetime of the unpackaged PSCs. Besides, the UV stability and lifetime of PSCs without a package are enhanced. We optimized the size of the ETP complexes to match the PSCs by optimizing the concentration. The ZnO/ETP device shows greatly enhanced photocurrent density and PCEs compared with the reference device.

## 2. Materials and Methods

### 2.1. Materials

The Photoactive materials PTB7-Th, PBDB-T-2F, and IT-4F were purchased from Solarmer Materials Inc. (Beijing, China). PC_71_BM was purchased from American Dye Sources Inc. Chlorobenzene (Sigma-Aldrich, St. Louis, MO, USA), 1,8-diiodoctane (DIO), and MoO_3_ were obtained from Sigma-Aldrich (St. Louis, MO, USA). Patterned ITO (indium tin oxide) glass substrates with a sheet resistance of 15 Ω/sq^−1^ was purchased from Shenzhen Display (Shenzhen, China). Zinc acetate dihydrate (Zn(Ac)_2_·2H_2_O, 99.99%), which is a precursor for ZnO, was bought from Sinopharm Chemical Reagent Co. (Shanghai, China). Ethanolamine (EA, 99%) was purchased from Aladdin Industrial Corporation (Beijing, China). Europium oxide (Eu_2_O_3_, 99.99%) was purchased from Sinopharm Chemical Reagent Co., Ltd., (Beijing, China). 1,10-phenanthroline (phen, 99%) and α-thenoyltrifluoroacetone (TTA, 99%) were obtained from Shanghai Darui Chemical Reagent Co., Ltd., (Shanghai, China). All reagents were used as received without further purification.

### 2.2. ZnO and Eu(TTA)_3_phen Complexes Synthesis

Zn(Ac)_2_·2H_2_O (2.195 g) was added into anhydrous ethanol (18.39 mL) in a round-bottom flask. After the turbid liquid was stable at 80 °C, EA (0.61 mL) was injected into the turbid reaction liquid, and the molar ratio of Zn(Ac)_2_·2H_2_O to EA was 1.0. After the addition of EA, the suspension immediately became clear. After stirring at a constant rate at 80 °C and refluxing for 2 h, a transparent ZnO precursor solution was obtained. After the reaction, it was cooled at room temperature for 3 h. The ZnO solution was diluted with absolute ethanol to obtain the desired concentration. Appropriate amounts of EuCl_3_, TTA, and phen were dissolved in ethanol, then added dropwise to the ZnO precursor solution with the molar ratio of 1:3:1, and then diluted with ethanol to the desired concentration. During this process, the ZnO/ETP complexes were formed. In addition, we kept the ratio of EuCl_3_, TTA, and phen unchanged, and added different volume of ethanol solution to the ZnO precursor solution to obtain different concentrations of ETP complexes. Herein, we renamed this ZnO/ETP solution in the form of ZnO/ETP1(*c* = 2 × 10^−3^ mol/L), ZnO/ETP2(*c* = 5 × 10^−3^ mol/L), and ZnO/ETP3(*c* = 8 × 10^−3^ mol/L), which represent different concentrations of ETP.

### 2.3. Device Fabrication

The device structure of PSCs (Figure 1) in this research is glass/ITO/ZnO/ETP/PTB7-Th–PC_71_BM/MoO_3_/Ag and glass/ITO/ZnO/ETP/PBDB-T-2F–IT-4F/MoO_3_/Ag. ITO substrates with conductivity of 15 Ω·sq^−1^ were ultrasonically cleaned for 15 min via ITO cleaning agent, deionized water, acetone, deionized water, and isopropyl alcohol, respectively, followed by UV-ozone treatment for 5 min. The ZnO precursor solution was spin-coated on the ITO conductive glasses at a spin-coating rate of 2000 rpm for 40 s and thermally annealed at 230 °C for 20 min to form a cathode buffer layer. The ITO substrates with the cathode buffer layer were then transferred into a nitrogen filled glovebox. PTB7-Th–PC_71_BM solutions (the ratio of donor/acceptor = 1:1.5, 9 mg·mL^−1^ polymer) were dissolved in chlorobenzene (CB) at 80 °C overnight with 3% of 1,8-diiodooctane (DIO) as an additive. The PTB7-Th–PC_71_BM blend solution was spin-coated on the cathode buffer layer at 1200 rpm for 60 s. PBDB-T-2F–IT-4F solutions (*D*/*A* = 1:1, 10 mg·mL^−1^ polymer) were dissolved in chlorobenzene at 40 °C overnight with 0.5% of 1,8-diiodooctane as an additive. The PBDB-T-2F–IT-4F blend solution was spin-coated on the cathode buffer layer at 3000 rpm for 40 s, and then annealed at 100 °C for 10 min. Subsequently, 10 nm MoO_3_ and 100 nm Ag electrodes were deposited onto the active layer by thermal evaporation in a vacuum chamber (<5 × 10^−4^ Pa).

### 2.4. Characterization

The current density–voltage (*J*–*V*) characteristics were recorded under AM 1.5 G solar illumination at an intensity of 100 mW cm^−2^ with Newport solar simulator by a Keithley 2420 source measurement (Shenzhen, China) in a nitrogen-filled glovebox. The external quantum efficiencies (EQEs) of inverted PSCs were analyzed by a certified Newport incident photon conversion efficiency (IPCE) measurement system (Newport, DE, USA). Photoluminescence (PL) of the ZnO/ETP was characterized by a CRAIC 20/30 PVTM microspectrophotometer (CRAIC, San Dimas, CA, USA). The transmittance and absorption spectra were performed by a Lambda 750 S (PerkinElmer, Waltham, MA, USA) spectrophotometer. X-ray photoelectron spectroscopy (XPS) data were measured with an ESCALAB250Xi electron spectrometer from Thermo (Waltham, USA) using 150 W Al Kα radiation. The surface morphology of the ZnO/ETP film was investigated by Quanta FEG 250 field emission Scanning Electron Microscope (SEM) (Carl Zeiss SMT, Oberkochen, Germany). The photo-induced force microscope (PiFM) used was a VistaScope from Molecular Vista, Inc. (San Jose, CA, USA). The stability of the PSCs was explored and the unencapsulated devices were continuously exposed to an ambient atmosphere. Fourier-transform infrared spectroscopy (FT-IR) data was measured with anNicolet 6700 Fourier transform infrared spectrometer from Thermo (Waltham, USA). High-resolution transmission electron microscopy (HRTEM) was performed using a FEI Talos F200i microscope (Thermo Fisher Scientific Inc., Waltham, MA, USA) operating at 200 kV. Thermo gravimetric analysis was measured with a SII TG/DTA 6300 (Waltham, MA, USA).

## 3. Results and Discussion

As shown in Figure 1a, the device structure of PSCs in this research is glass/ITO/ZnO/ETP/PTB7-Th–PC_71_BM/MoO_3_/Ag. The visible light can be transmitted from the ITO glass substrate and ZnO/ETP layer and absorbed by the active layer materials. After the UV light passes through the glass substrate, it is absorbed by the ZnO/ETP layer and re-emits visible light to be absorbed by the active layer materials. The absorption spectra of the neat PTB7-Th and PC_71_BM films and PL spectra of ZnO and ZnO/ETP are shown in Figure 1c, ZnO has weak fluorescence at 420–675 nm, there is no obvious peak at 612 nm, and the ETP complexes are excited by UV light and re-emits red light (612 nm), indicating that ETP is the source of emission. In addition, it is matched well with the absorption of the donor material (PTB7-Th), thus helping to enhance photovoltaic performances in PSCs. The photo of ZnO/ETP film under UV light is also shown in Figure 1c, and it is obviously that the ZnO/ETP film spin-coated on ITO substrate emitted the red light.

X-ray photoelectron spectroscopy (XPS) reveals the composition of the prepared ZnO and ZnO/ETP films. As shown in Figure 2a,b, it can be seen that the survey spectra of Zno and ZnO/ETP films characterize the peak of Zn, O, and C elements, and the binding energy can be calibrated by C 1s (284.8 eV). The ETP complexes are introduced into the ZnO film, and because the Eu atom and the organic ligands form coordinate bonds in the complexes, the peak of the Eu element appears in the measurement spectrum of Figure 2b. In Figure 2c,d, the binding energy of the Zn 2p_1/2_ and Zn 2p_2/3_ peaks in ZnO/ETP film has no significant difference from the binding energy of the Zn 2p_1/2_ and Zn 2p_2/3_ peaks in Zno film [34]. Therefore, Zn is less likely to form bonds with organic ligands or be substituted by Eu. Instead, the organic ligands fill the oxygen vacancy in ZnO film. The O 1s core level spectra of ZnO and ZnO/ETP films are shown in Figure 2e,f, respectively. In Figure 2e, there are two peaks in the level spectra: the Oa peak at 530.3 eV is associated with the lattice oxygen atoms, which is the binding energy of the Zn–O bond; the Ob peak at 532 eV usually corresponds to oxygen-deficient defects, such as oxygen vacancies [35,36]. It is obviously that the Ob peak of oxygen-deficient defects in the ZnO/ETP level spectra is decreased relative to the Ob peak of ZnO level spectra. This indicates that the incorporation of organic ligands into ZnO can effectively reduce the oxygen deficiency defects of the film, thus we conclude that the reduction in defect density is the result of the physical absorption of O atoms from TTA filling up the surface oxygen vacancies of the ZnO film. The higher surface defect densities introduce different intragap levels. These interstitial states become the recombination centers for photogenerated charge carriers, resulting in significant charge carrier recombination and reduced charge selectivity at the cathode contact and resulting in a low fill factor [35]. By filling the oxygen vacancies of the ZnO film, the recombination of photogenerated charge carriers can be effectively reduced, and the fill factor can be improved.

As has been discussed, the nanostructure morphology of ETP plays an important role in determining the photovoltaic performances of PSCs, because big aggregation of ETP will destroy the hyperfine structure of PSCs and cause hole and electron recombination. In this work, SEM and PiFM were utilized to characterize the morphology of the ZnO/ETP blend film.

In Figure 3a,b, we use SEM to investigate the surface morphology of ZnO film and ZnO/ETP blend film spin-coated on the ITO substrates. The SEM image of pure ZnO is a homogeneous film. However, from the SEM image of the ZnO/ETP blend film, it can be observed that some of the smaller dark regions are evenly distributed in the light region. Figure 3c,d provide the topography and photo-induced force microscopy (PiFM) image of the ZnO/ETP blend film, respectively.

The HR-TEM (High-Resolution Transmission Electron Microscope) image of ZnO/ETP film scraping from a glass substrate and the corresponding elemental mappings of O, F, S, Eu, and Zn are shown in Figure 5b–h; elements such as F, S, and Eu in the complexes are uniformly distributed in the ZnO/ETP film, and the peaks of F, S, of Eu can also be found in the corresponding EDS (Energy Dispersive Spectrometer) spectra of the ZnO/ETP film in Figure 5a. Figure 3c shows the AFM (Atomic Force Microscope) topography image of the ZnO/ETP film. As results, it is observed in the AFM image that the height difference of the surface of ZnO/ETP film is less than 6.10 nm, which indicates that the surface of the ZnO/ETP film is relatively flat and there is no large aggregate. Therefore, we believe that ETP complexes have no large agglomeration in the ZnO film and have good dispersibility. It is obvious that the conventional atomic force microscopy (AFM) images cannot accurately reflect the chemical space information. PiFM combines AFM with the analytical capability of infrared laser to demonstrate the spatial distribution of each chemical composition. As shown in Figure 4, ZnO and ZnO/ETP have characteristic FTIR (Fourier transform infrared) absorption peaks, related to their chemical structure. In Figure 3d, PiFM by imaging at the characteristic FTIR wavelengths correspond to absorption peaks of ETP (1142 cm^−1^). The PiFM image shows the spatially mapped patterns of the ZnO/ETP film are clearly visible and the yellow signal represents the phase distribution of ETP complexes [37,38], which are evenly dispersed in the film. The PiFM image is consistent with the SEM image. The uniform dispersion of ETP complexes in ZnO film is critical for the photovoltaic performance of the PSCs; these results demonstrate that the distribution of ETP complexes in the ZnO film is relatively uniform. The TEM images of the ZnO/ETP precursor solution are shown in Figure 5i–k; the size and distribution of the ETP complexes can be controlled by changing the concentration of the complexes. The uniformly distributed ETP complexes absorb ultraviolet light more widely and re-emit visible light to be absorbed by the active layer, which more effectively broadens the spectral response range of the active layer, and enhances the photovoltaic performance of the PSCs.

Figure 6b shows the optical transparency spectra of the ITO, ZnO, and ZnO/ETP films. The ZnO and ZnO/ETP films were spin-coated on the ITO substrates. It is found that the transmittance of ZnO/ETP film is lower than the ITO and ZnO films in the region of 330–380 nm, which is consistent with the spectral response range of ETP complexes, and the ETP complexes are excited in the UV region and re-emit visible light to be absorbed by the active layer materials. Besides, the ZnO/ETP film shows good transmittance performance in the visible region. The ZnO/ETP film shows better transmittance than ITO in the region of 400–500 nm, and the transmittance of ZnO/ETP and ITO films in the region of 500–800 nm are substantially the same, and do not affect the absorption of the active layer materials. Thus, the ZnO/ETP film can effectively reduce the transmission of UV light and UV light irradiation of active layer materials, and decrease the defects produced by the polymer under the induction of UV light and the degradation of the device. The ZnO/ETP layer will not significantly affect the light absorption of the active layer, thus it can improve the UV stability of the device of PSCs.

We varied the annealing temperature and concentrations of the ZnO/ETP layer to investigate its influence on device performance and achieve the optimal device performance. The PSCs parameters of different annealing temperature of ZnO and ZnO/ETP as cathode buffer layers are shown in Table 1. We measured the conductivities of ZnO and ZnO/ETP at different annealing temperatures and found that the conductivity of ZnO and ZnO/ETP was basically the same at the same temperature. Especially, the electrical conductivity of ZnO is 0.098 S/cm at 230 °C. We can find the optimal photovoltaic performance of PSCs when the annealing temperature of the cathode buffer layers is 230 °C. The photovoltaic parameters of the ZnO/ETP device including *J_sc_*, *FF* (Fill factor), and PCEs are superior to the ZnO device. In addition, the annealing temperature of 230 °C and a short annealing time do not decompose the ETP complexes. As shown in Figure 6a, we measured the TGA (Thermogravimetric Analysis) spectra of ETP complexes. High temperature heat treatment of ETP complexes for more than 20 min only causes less than 5 percent decomposition, which does not significantly affect the photovoltaic performance of PSCs based on ZnO/ETP.

The optimal current density–voltage (*J*–*V*) curves characteristics under the illumination of AM 1.5 G (100 mW·cm^−2^) of the devices are presented in Figure 7a,c, and the PSCs parameters are shown in Table 2. The PTB7-Th–PC_71_BM device based on the pure ZnO interlayer showed efficiency of 8.11% with a *V_oc_* of 0.77 V, *J_sc_* of 16.77 mA cm^−2^, and *FF* of 62.22%. We used different concentrations of ETP mixed with ZnO as the electronic transport layer, and both of them exhibited various degrees of enhancement on device performances. The optimal concentration of ETP is determined to be 0.005 mol/L, and a *V_oc_* of 0.77 V, *J_sc_* of 17.64 mA cm^−2^, and *FF* of 68.13% are obtained, resulting in a higher PCE of 9.22%, which is better than the reference device—a 13.7% increase compared with the devices with the pure ZnO interlayer. A higher photocurrent density was achieved in the ZnO/ETP devices than in the ZnO reference device, likely because of its wider spectral response range than that of the ZnO device. In addition, the organic complexes fill the oxygen vacancies of the ZnO film, so the value of *FF* increases from 62.22% to 68.13%, which we will describe later. In addition, the PBDB-T-2F–IT-4F device based on ZnO/ETP2 can obtain PCEs of 13.12%, which indicates the down-conversion materials (ETP complexes) can improve the spectral response range and enhance the photovoltaic performance of PSCs, and can be applied to the devices based on different polymers. To verify the high photocurrent density of ZnO/ETP devices, we measured the external quantum efficiency (EQE). Figure 7b,d show the EQE spectra of the PSCs based on the pure ZnO and ZnO/ETP interlayer. The *J_sc_* of PTB7-Th–PC_71_BM devices calculated from the EQE spectra are 16.43 mA cm^−2^ (2.06%), 16.60 mA cm^−2^ (1.27%), 17.49 mA cm^−2^ (0.86%), and 17.45 mA cm^−2^ (0.80%) for the device with ZnO, ZnO/ETP1, ZnO/ETP2, and ZnO/ETP3, respectively. Compared with the pure ZnO cathode transport layer devices, the ZnO/ETP interlayer-based devices have significantly enhanced EQE in the 365–705 nm wavelength range, indicating that the ZnO/ETP device has a lager *J_sc_*, which is in accordance with the measured photocurrent density. More importantly, in the case of the incorporation of ETP complexes, the enhancement of EQE is mainly in the wavelength range of 350–500 nm, as the ETP complexes are excited in this range and re-emit red light, which matches the absorption peak of the donor material PTB7-Th, thereby increasing the photocurrent density of PSCs. In summary, the ETP down-conversion material can effectively enhance the photoelectronic response of PSCs, extending the spectral response range of the PSCs to the UV region, resulting in higher *J_sc_*, and the EQE clearly correlates the improvement in spectral response range to lanthanide luminescence.

To further investigate the light absorption and charge collection process, photocurrent density (*J_ph_*) and effective voltage (*V_eff_*) for the ZnO and ZnO/ETP device are shown in Figure 8a. Generally, *J_ph_* is defined as the following formula: *J_ph_* = *J_L_* − *J_D_*, for which *J_L_* and *J_D_* are the photocurrent densities under illumination and dark conditions, respectively. *V_eff_* is defined as the following formula: *V_eff_* = *V*_0_ − *V_a_*, where *V*_0_ is the voltage, *J_ph_* = 0, and *V_a_* is the applied external voltage bias [39,40]. Apparently, the value of *J_ph_* for the ZnO and ZnO/ETP devices linearly increased at a lower value of *V_eff_* and tended to saturate (*J_sat_*) when *V_eff_* = 2 V. It is obvious that the *J_sat_* value of the ZnO/ETP device is higher than that of the ZnO device; the enhanced *J_sat_* indicates that incorporating the ETP complexes can enhance the photocurrent of PSCs. We use the maximum exciton generation rate (*G*_max_) to report the light absorption of the devices. *G*_max_ is defined as the following formula: *J_sat_* = *qG*_max_*L*, where *q* is the electronic charge and *L* is the thickness of the active layer [41,42]. Obviously, the *G*_max_ value of the ZnO/ETP device is higher than that of the ZnO device when the thickness of the active layer is constant. The increase in *G*_max_ is the result of enhanced light absorption of the ZnO/ETP device. Therefore, the introduction of ETP complexes in ZnO can effectively enhance the light absorption of PSCs and improve its photovoltaic performance.

The *J_sc_* was measured as a function of the light intensity (*P*_light_) for the ZnO and ZnO/ETP devices to investigate the main mechanism of bimolecular recombination in the PSCs. In the logarithmic plot of Figure 8b, the *J_sc_* as a function displays a power–law dependence on *P*_light_ (*J_sc_*∝$P_{\mathrm{light}}^{S}$), where the fitted *S* value close to 1 reflects weak bimolecular recombination [43,44]. The slopes *S* for ZnO and ZnO/ETP devices are 0.982 and 0.990, respectively. It is obviously that the *S* value of the ZnO/ETP device is closer to 1, suggesting the weaker bimolecular recombination in PSCs. The main reason for charge loss in the active layer should be bimolecular recombination, so it can well explain the improvement of the fill factor of the ZnO/ETP device (68.13%) compared with the ZnO device (62.22%).

Device stability is another major factor in determining the applicability of PSCs for industrial production in addition to high efficiency. However, previous studies have shown that these devices degrade in a few hours under continuous illumination [45]. UV light irradiation is the major factor of the degradation of PSCs; the C–H and C=C bonds of conjugated polymers in the active layer are broken under the induction of UV light to produce defects [46]. In order to investigate the stability difference between the ZnO and ZnO/ETP device under solar light, we studied the PCEs decay curves of the ZnO and ZnO/ETP device stored for 17 days in an N_2_-filled glove box. In Figure 9, the ZnO/ETP solar cell presents a slower normalized efficiency reduction and is measured for 17 days. Owing to the degradation of UV light, the efficiency of the reference solar cell decreased to 0 after 10 days. Relatively slow efficiency decay can be achieved by mixing the ETP complexes with ZnO to form a cathode transport layer to inhibit UV light illumination to inhibit photo degradation of PSCs; the ZnO/ETP device still had a normalized efficiency of 65% when the reference device decreased to 0. In summary, ETP complexes can enhance the power conversion efficiency while prolonging the lifetime of PSCs.

Figure 10 is the mechanism of UV light energy transfer to PTB7-Th. It can be seen from the absorption spectra of ETP that there is a strong absorption band in 200–400 nm and no obvious absorption in the visible light region. Therefore, the ETP complexes do not affect the absorption of visible light by the active materials. The UV light can be absorbed by ETP complexes when it is irradiated on ZnO/ETP film, which effectively reduces the influence of UV light on the active materials and prolongs the lifetime of the device. This down-conversion material (ETP) absorbs UV light and re-emits red light (peak at 612 nm). In particularly, PTB7-Th has a strong absorption at 550–750 nm, and PBDB-T-2F has a strong absorption at 400–700nm.The absorption spectra of these two donor materials match well with the red light (612 nm) re-emitted by the ETP complexes, so this red light can be absorbed by the active materials. Therefore, the ETP down-conversion material can effectively enhance the photoelectronic response of PSCs, extending the spectral response range of the PSCs to the UV region.

## 4. Conclusions

In conclusion, we have presented an efficient approach to enhance the photovoltaic performance and UV stability of PSCs by introduction of an Eu(TTA)_3_phen lanthanide down-conversion material to the cathode buffer layer. By this method, the nanostructure of Eu(TTA)_3_phen can be well controlled to avoid destroying the hyperfine structure of PSCs. The ZnO/ETP layer can absorb the UV light and re-emit it at the visible region, and the emitted light matches well with the absorption of active polymer PTB7-Th and PBDB-T-2F; therefore, the spectral response range of the PSCs can be extended to the UV region. We obtained the best device performance when the concentration of the ETP complexes was 5 × 10^−3^ mol/L, and the *J_sc_*, *V_oc_*, and *FF* of the ZnO/ETP2 device were 17.64 mA cm^−2^, 0.77 V, and 68.13%, respectively, resulting in a higher PCE of 9.22%, increased by 13.7% when compared with the ZnO device. Moreover, the ZnO/ETP cathode transport layer in PSCs can prolong the lifetime of PSCs by absorbing the UV light to inhibit the UV degradation of the active layer material. Therefore, this research might provide a promising avenue for high-performance polymer solar cells’ designs.

## Figures and Tables

**Figure 1 nanomaterials-10-00080-f001:**
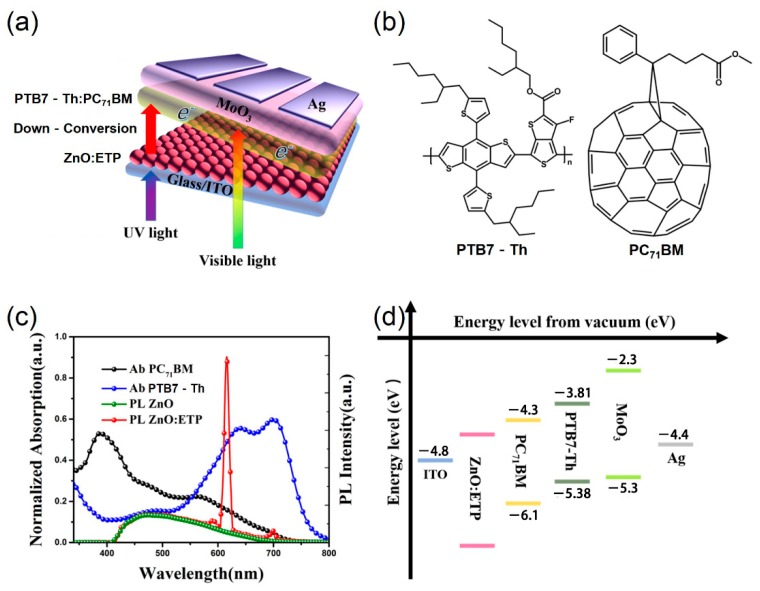
(**a**) Schematic illustration of the device architecture in this work. (**b**) The molecular structure of polymer PTB7-Th and PC_71_BM. (**c**) Absorption spectra of the neat PTB7-Th and PC_71_BM films and photoluminescence (PL) spectra of ZnO and ZnO/Eu(TTA)_3_phen (ETP). (**d**) Energy level diagram of the components of the device.

**Figure 2 nanomaterials-10-00080-f002:**
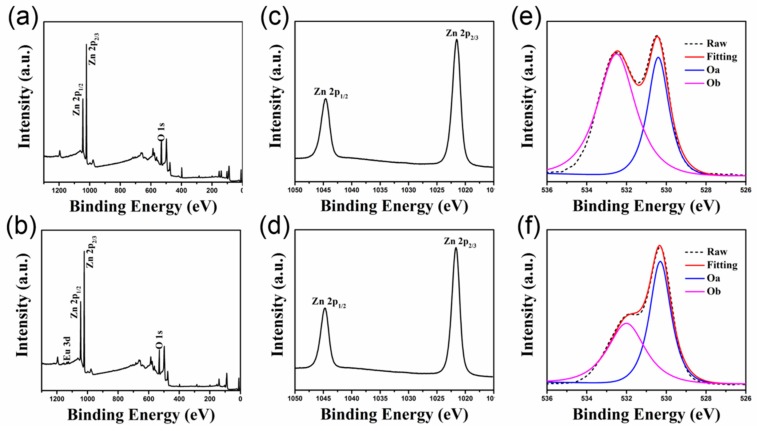
X-ray photoelectron spectroscopy (XPS) spectra of ZnO and ZnO/ETP films (**a**,**b**) survey scan, (**c**,**d**) Zn 2p spectra, and (**e**,**f**) O 1s spectra of ZnO and ZnO/ETP films.

**Figure 3 nanomaterials-10-00080-f003:**
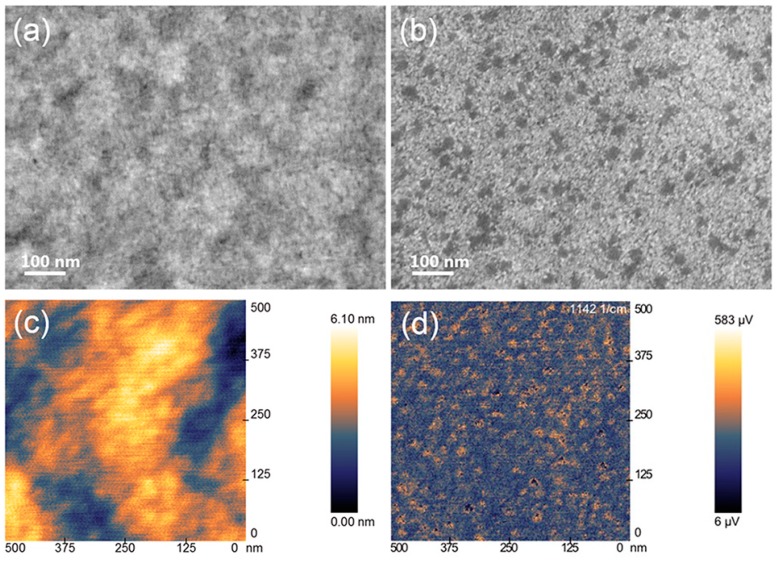
SEM images of (**a**) ZnO film and (**b**) ZnO/ETP blend film, (**c**) AFM topography image, and (**d**) photo-induced force microscopy (PiFM) image of ZnO/ETP blend film.

**Figure 4 nanomaterials-10-00080-f004:**
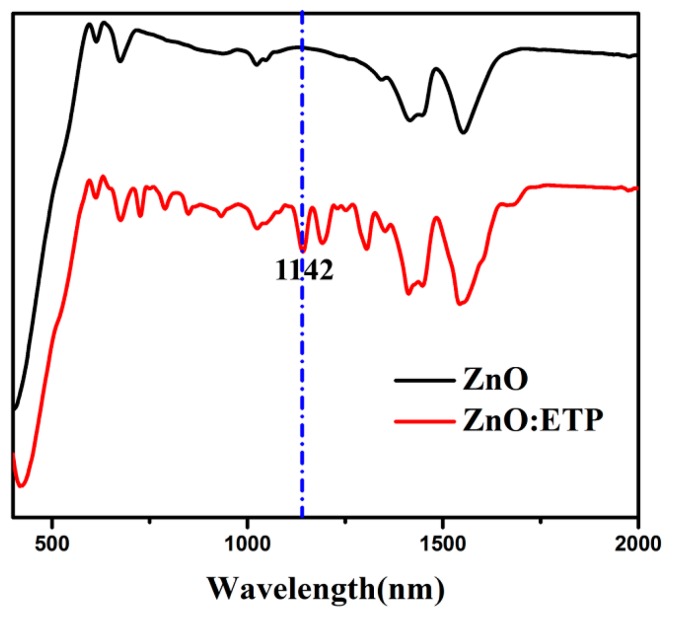
The Fourier transform infrared (FTIR) absorption peaks of ZnO and ZnO/ETP.

**Figure 5 nanomaterials-10-00080-f005:**
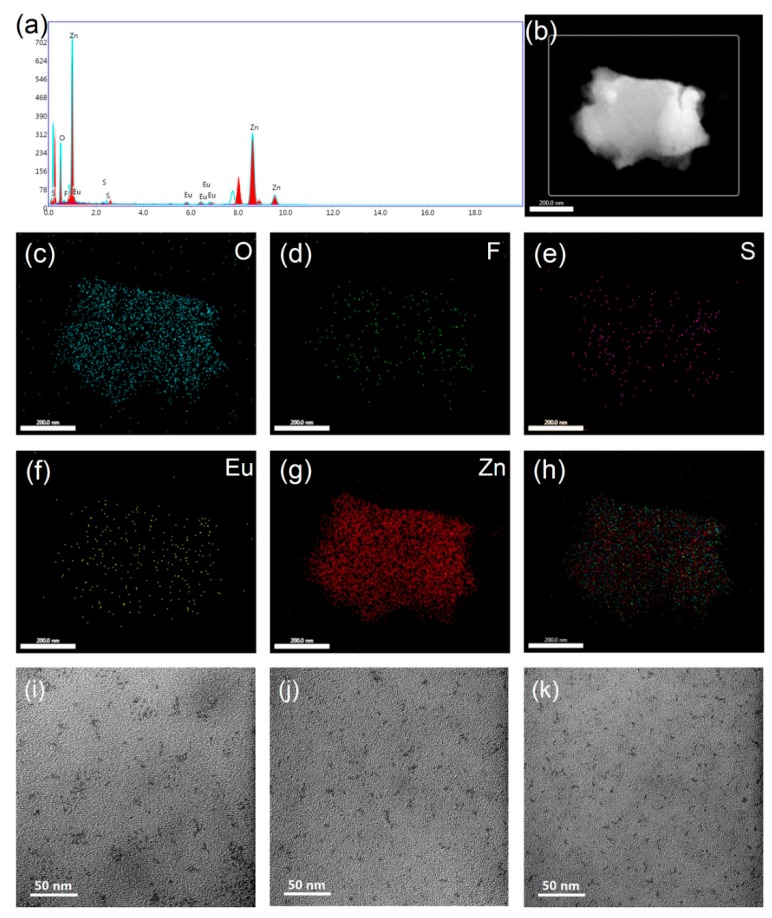
(**a**) The corresponding EDS spectra of ZnO/ETP film; HR-TEM image of (**b**) ZnO/ETP film scraping from a glass substrate and (**c**–**g**) the corresponding elemental mappings of O, F, S, Eu, and Zn, respectively; and (**i**–**k**) TEM images of ETP1, ETP2, and ETP3 precursor solution, respectively.

**Figure 6 nanomaterials-10-00080-f006:**
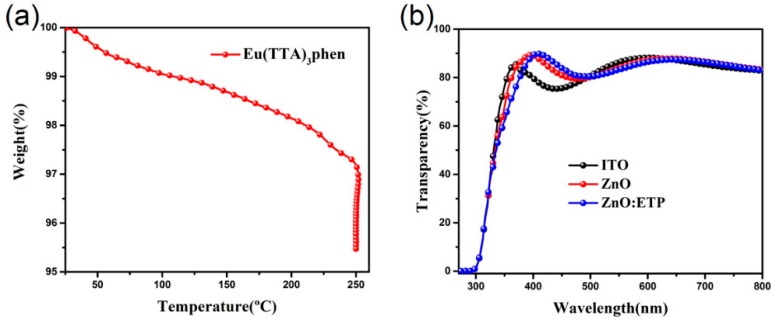
(**a**) TGA spectra of ETP complexes, heating rate (15 °C/min), 250 °C for 20 min; (**b**) transparency spectra of ITO, intrinsic ZnO, and ZnO/ETP films.

**Figure 7 nanomaterials-10-00080-f007:**
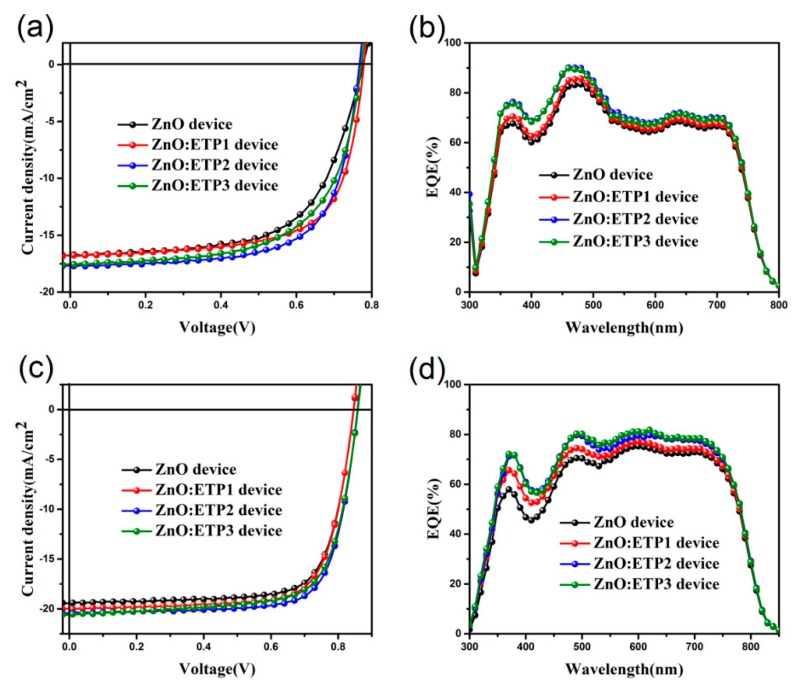
(**a**,**c**) Current density versus voltage characteristics; (**b**,**d**) external quantum efficiency (EQE) curves of devices based on PTB7-Th–PC_71_BM and PBDB-T-2F–IT-4F using ZnO and ZnO/ETP (different concentrations) as cathode buffer layers.

**Figure 8 nanomaterials-10-00080-f008:**
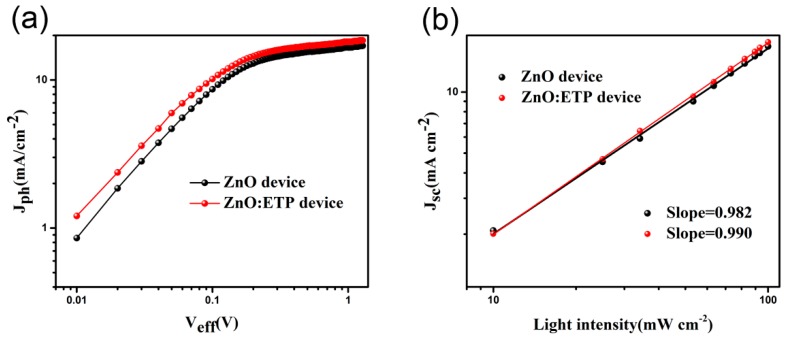
(**a**) Photocurrent density versus effective voltage curves under AM1.5 G illumination at 100 mW cm^−2^; (**b**) light intensity dependence of *J_sc_* of the devices processed with ZnO and ZnO/ETP.

**Figure 9 nanomaterials-10-00080-f009:**
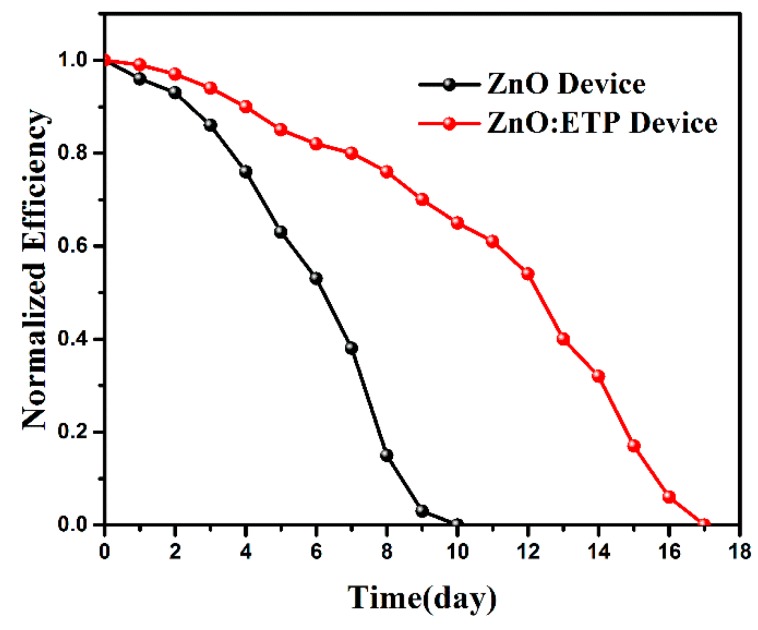
Power conversion efficiency (PCE) decay curves of the ZnO and ZnO/ETP device (based on PTB7-Th–PC_71_BM) stored for 17 days in an N_2_-filled glove box.

**Figure 10 nanomaterials-10-00080-f010:**
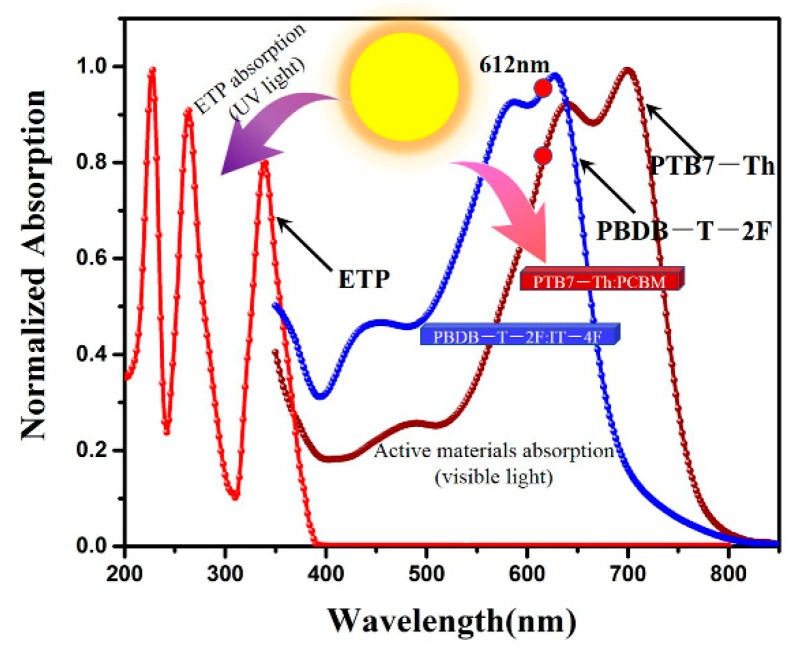
UV/visible absorption spectra of ETP, PTB7-Th–PC_71_BM, and PBDB-T-2F–IT-4F (the red circle symbolizes the peak position in the ETP emission spectrum; the arrows symbolize that UV light and visible light are absorbed by the ETP and active layer materials, respectively).

**Table 1 nanomaterials-10-00080-t001:** Photovoltaic performance parameters of the devices based on PTB7-Th–PC_71_BM blends, ZnO, and ZnO/Eu(TTA)_3_phen (ETP) (*c* = 4 × 10^−3^ mol/L) treated with different annealing temperatures as cathode buffer layers under the illumination of AM 1.5 G, 100 mW cm^−2^. PCE, power conversion efficiency.

Temperature (°C)	Devices	Conductivity (S/cm)	*V*_oc_ (V)	*J*_sc_ (mA cm^−2^)	*FF* (%)	PCE (%)
170	ZnO	0.034	0.77 (0.76 ± 0.01)	16.20 (16.00 ± 0.20)	61.75 (61.16 ± 0.59)	7.71 (7.50 ± 0.21)
	ZnO/ETP	0.031	0.76( 0.75 ± 0.01)	16.69 (16.50 ± 0.19)	60.11 (59.55 ± 0.56)	7.72 (7.53 ± 0.19)
200	ZnO	0.058	0.76 (0.75 ± 0.01)	16.74 (16.55 ± 0.19)	62.62 (62.04 ± 0.58)	7.94 (7.74 ± 0.20)
	ZnO/ETP	0.062	0.77 (0.76 ± 0.01)	16.59 (16.43 ± 0.16)	66.33 (65.81 ± 0.52)	8.46 (8.27 ± 0.19)
230	ZnO	0.098	0.77 (0.76 ± 0.01)	16.61 (16.42 ± 0.19)	63.03 (62.45 ± 0.58)	8.09 (7.91 ± 0.18)
	ZnO/ETP	0.095	0.77 (0.76 ± 0.01)	17.63 (17.47 ± 0.16)	66.06 (65.49 ± 0.51)	9.01 (8.85 ± 0.16)
260	ZnO	0.091	0.79 (0.78 ± 0.01)	16.71(16.53 ± 0.18)	62.89 (62.30 ± 0.59)	8.25 (8.07 ± 0.18)
	ZnO/ETP	0.091	0.77 (0.76 ± 0.01)	16.62 (16.42 ± 0.18)	68.16 (67.63 ± 0.53)	8.78 (8.61 ± 0.17)

**Table 2 nanomaterials-10-00080-t002:** Photovoltaic performance parameters of the devices based on PTB7-Th–PC_71_BM and PBDB-T-2F–IT-4F using ZnO and ZnO/ETP (different concentrations) as cathode buffer layers under the illumination of AM (air mass) 1.5 G, 100 mW cm^−2^.

Polymer	Devices	*V_oc_* (V)	*J_sc_* (mA cm^−2^)	*FF* (%)	PCE (%)
PTB7-Th–PC_71_BM	ZnO	0.77 (0.76 ± 0.01)	16.77 (16.54 ± 0.23)	62.22 (61.62 ± 0.60)	8.11 (8.00 ± 0.21)
	ZnO/ETP1	0.78 (0.77 ± 0.01)	16.81 (16.60 ± 0.21)	67.99 (67.40 ± 0.59)	8.90 (8.71 ± 0.19)
	ZnO/ETP2	0.77 (0.76 ± 0.01)	17.64 (17.48 ± 0.16)	68.13 (67.61 ± 0.52)	9.22 (9.07 ± 0.15)
	ZnO/ETP3	0.77(0.76 ± 0.01)	17.59 (17.42 ± 0.17)	63.03 (62.46 ± 0.57)	8.57 (8.40 ± 0.17)
PBDB-T-2F–IT-4F	ZnO	0.85 (0.84 ± 0.01)	19.43 (19.12 ± 0.31)	74.06 (73.41 ± 0.65)	12.17 (11.88 ± 0.29)
	ZnO/ETP1	0.85 (0.84 ± 0.01)	19.92 (19.63 ± 0.29)	74.00 (73.37 ± 0.63)	12.47 (12.20 ± 0.27)
	ZnO/ETP2	0.86 (0.85 ± 0.01)	20.39 (20.14 ± 0.25)	74.98 (74.37 ± 0.61)	13.12 (12.87 ± 0.25)
	ZnO/ETP3	0.86 (0.85 ± 0.01)	20.60 (20.33 ± 0.27)	71.81 (71.20 ± 0.61)	12.17 (11.91 ± 0.26)

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
