# Peer review of "Hybrid ZnO Electron Transport Layer by Down Conversion Complexes for Dual Improvements of Photovoltaic and Stable Performances in Polymer Solar Cells"

_nanomaterials, 2020, doi:10.3390/nano10010080_

Round 1
Reviewer 1 Report
1) According to the published papers, the Voc of PTB7-Th:PC71BM based solar cell is 0.81~0.83 [V] (Adv. Funct. Mater. 26, 6635, 2016; Nature Photonics. 9, 174, 2015). However, in this paper the Voc is about ~0.77 [V]. Why is the Voc decreased? It is recommended to add a description of this part in the manuscript.
2) ZnO and ZnO:ETP was formed by the sol-gel method. Usually, a high temperature heat treatment is necessary about 400°C, in which ZnO thin film was form with high conductivity. In this paper, the heat treatment was performed at 170~260 degrees. What is the conductivity of ZnO thin film formed at this temperature? It is recommended to add a description of this part in the manuscript.
3) In the case of PTB7-Th: PC71BM, the FF of ZnO:ETP is greatly improved, but not in the case of PBDB-T-2F:IT-4F. What is the reason?
Author Response
Question 1: According to the published papers, the Voc of PTB7-Th:PC71BM based solar cell is 0.81~0.83 [V] (Adv. Funct. Mater. 26, 6635, 2016; Nature Photonics. 9, 174, 2015). However, in this paper the Voc is about ~0.77 [V]. Why is the Voc decreased? It is recommended to add a description of this part in the manuscript.
Answer: Thank you very much for this good suggestion. The reason the voltage drops is due to the inorganic electron-collecting interlayer could have poor interfacial contact with the organic active layer, resulting in poor electron extraction. Although the device structures of glass/ITO/PFN/PTB7-Th:PC71BM/MoO3/Ag and glass/ITO/PEDOT:PSS/PTB7-Th:PC71BM/PFN/Al have higher open circuit voltage and photoelectric conversion efficiency, but the PFN and PEDOT: PSS is acidic (need to add acetic acid during PFN preparation) and has a corrosive effect on ITO, both of which are detrimental to device lifetime[28,29]. Therefore, in order to avoid this problem and improve the stability of the device, we use ZnO as the cathode buffer layer of the solar cells. We have added instructions in Page 2(line 83) for this section to the manuscript and added two related citations.
Liao, S.H.; Jhuo, H.-J.; Cheng, Y.-S.; Chen, S.-A. Fullerene Derivative-Doped Zinc Oxide Nanofilm as the Cathode of Inverted Polymer Solar Cells with Low-Bandgap Polymer (PTB7-Th) for High Performance. Adv. Mater. 25, 4766-4771. He, Z.; Zhong, C.; Su, S.; Xu, M.; Wu, H.; Cao, Y. Enhanced power-conversion efficiency in polymer solar cells using an inverted device structure. Nat. Photonics 2012, 6, 593-597.
Question 2: ZnO and ZnO:ETP was formed by the sol-gel method. Usually, a high temperature heat treatment is necessary about 400°C, in which ZnO thin film was form with high conductivity. In this paper, the heat treatment was performed at 170~260 degrees. What is the conductivity of ZnO thin film formed at this temperature? It is recommended to add a description of this part in the manuscript.
Answer: Thank you very much for the kind suggestions. The annealing temperature of ZnO between 120-300 degrees Celsius is common in the application of polymer solar cells. Low temperature processing is an inevitable requirement for the development of glass substrates to flexible devices. We measured the conductivities for ZnO and ZnO:ETP using the four-probe method and add the values to Table 1. The description in Page 10(line 269) is as follow in the revised manuscript:
We measured the conductivities of ZnO and ZnO:ETP at different annealing temperatures and found that the conductivity of ZnO and ZnO:ETP were basically the same at the same temperature. Especially, the electrical conductivity of ZnO is 0.098 S/cm at 230 °C.
Table 1. Photovoltaic performance parameters of the devices based on PTB7-Th:PC71BM blends, ZnO and ZnO:ETP (c=4 × 10-3 mol/L) treated with different annealing temperatures as cathode buffer layers under the illumination of AM 1.5G, 100 mW cm-2.
|
Temperature(°C) |
Devices |
Conductivity (S/cm) |
Voc (V) |
Jsc (mA cm-2) |
FF (%) |
PCE (%) |
|
170 |
ZnO |
0.034 |
0.77 (0.76±0.01) |
16.20 (16.00±0.20) |
61.75 (61.16±0.59) |
7.71 (7.50±0.21) |
|
ZnO:ETP |
0.031 |
0.76( 0.75±0.01) |
16.69 (16.50±0.19) |
60.11 (59.55±0.56) |
7.72 (7.53±0.19) |
|
|
200 |
ZnO |
0.058 |
0.76 (0.75±0.01) |
16.74 (16.55±0.19) |
62.62 (62.04±0.58) |
7.94 (7.74±0.20) |
|
ZnO:ETP |
0.062 |
0.77 (0.76±0.01) |
16.59 (16.43±0.16) |
66.33 (65.81±0.52) |
8.46 (8.27±0.19) |
|
|
230 |
ZnO |
0.098 |
0.77 (0.76±0.01) |
16.61 (16.42±0.19) |
63.03 (62.45±0.58) |
8.09 (7.91±0.18) |
|
ZnO:ETP |
0.095 |
0.77 (0.76±0.01) |
17.63(17.47±0.16) |
66.06 (65.49±0.51) |
9.01 (8.85±0.16) |
|
|
260 |
ZnO |
0.091 |
0.79 (0.78±0.01) |
16.71(16.53±0.18) |
62.89 (62.30±0.59) |
8.25 (8.07±0.18) |
|
ZnO:ETP |
0.091 |
0.77 (0.76±0.01) |
16.62(16.42±0.18) |
68.16 (67.63±0.53) |
8.78 (8.61±0.17) |
Question 3: In the case of PTB7-Th:PC71BM, the FF of ZnO:ETP is greatly improved, but not in the case of PBDB-T-2F:IT-4F. What is the reason?
Answer: This is a very good question. As the PBDB-T-2F: IT-4F-based devices have a relatively mature manufacturing process, the value of FF remains at a high level. Therefore, the improvement of the performance of the electron transport layer in this study did not significantly increase the value of FF.

Reviewer 2 Report
The authors present a study of hybrid ZnO electron transport layers with down conversion complexes. The benefit for two different types of polymer solar cells is demonstrated. Power conversion efficiencies are improved and higher stability of the devices is claimed.
1.
It is not clear for what kind of devices (PTB7-Th or PBDB-T) the long term study in Figure 9 has been carried out. As no error bars are shown for the measurements I conclude that only one type of device has been investigated. The has to be specified. This refers also to the abstract (line 26). Of course it would be best to add the long term experiment for the second type of polymer solar cells. If this is not possible one should be more careful with a general conclusion.
2.
In line 75 it is referred to previous work with an increase of PCEs by 13.7% without citation. However, the increase by 13.7% is also the claim for this publication (line 271), so rephrasing seems to be necessary.
3.
The scheme in Figure 1 is exclusively for the PTB7-Th type of device. Why is the description of the second type of device (PBDB-T) missing?
4.
Concerning the degradation of polymer solar cells also the influence of morphological degradation has to be mentioned. This has been demonstrated in a number of studies including the PTB7-Th:PC71BM system with DIO as a solvent additive:
https://doi.org/10.1021/acsenergylett.8b02311
But again, the kind of device measured and presented in the plot of Figure 9 has to be specified.
5.
The caption of Figure 10 the meaning of arrows and red circles should be given. Although both systems are put into the figure, the discussion is reduced to the PTB7-Th system (line 337).
Author Response
The authors present a study of hybrid ZnO electron transport layers with down conversion complexes. The benefit for two different types of polymer solar cells is demonstrated. Power conversion efficiencies are improved and higher stability of the devices is claimed.
Question 1: It is not clear for what kind of devices (PTB7-Th or PBDB-T-2F) the long term study in Figure 9 has been carried out. As no error bars are shown for the measurements I conclude that only one type of device has been investigated. The has to be specified. This refers also to the abstract (line 26). Of course it would be best to add the long term experiment for the second type of polymer solar cells. If this is not possible one should be more careful with a general conclusion.
Answer: Thank you very much for the kind suggestions. In this work, we mainly studied the effect of ZnO: ETP interlayer on the photovoltaic performance and stability of PSCs. In the study of PBDB-T-2F: IT-4F devices, due to the emission spectrum of ETP complexes and the absorption spectra of PBDB-T-2F can be well matched, we mainly discuss the improvement of photovoltaic performance of ZnO:ETP devices as a supplementary explanation. We apologize for the inaccuracy of some of the statements in the manuscript, and have modified them in Page 1(line 25) of the manuscript:
Besides, in the research on PTB7-Th:PC71BM device, the stability of the device based on ZnO:ETP layer is prolonged by 70% compared to the ZnO device. The results suggest that the ZnO:ETP layer plays the role of enhanced photovoltaic performance and prolonged device stability as well as reducing photo-loss and UV degradation for PSCs.
Figure 9. PCEs decay curves of the ZnO and ZnO:ETP device(based on PTB7-Th:PC71BM) stored in 17 days in N2 filled glove box.
Question 2: In line 75 it is referred to previous work with an increase of PCEs by 13.7% without citation. However, the increase by 13.7% is also the claim for this publication (line 271), so rephrasing seems to be necessary.
Answer: Thank you very much for the kind suggestions. We are very sorry for this mistake, we have corrected the relevant description in Page 2(line 78) of the manuscript and adjusted the structure of the introduction appropriately. The corrections in the manuscript are as follows:
Typically, Eu(TTA)3phen (ETP) luminescent complexes have irregular structures and large size differences, which are not ideal when applied to a precise structure of PSCs. Hence, it is necessary to adjust the size and distribution of ETP complexes by changing the concentration of the complexes to optimize the photovoltaic performance of PSCs.
In this work, we constructed simple and effective way to regulated the size of solid micelles of rare earth complexes and then applied them to polymer solar cells to increase the photoelectric conversion efficiency.
Question 3: The scheme in Figure 1 is exclusively for the PTB7-Th type of device. Why is the description of the second type of device (PBDB-T) missing?
Answer: This is a very good question. In this work, we mainly studied the effect of ZnO: ETP interlayer on the photovoltaic performance and stability of PSCs. The PBDB-T-2F: IT-4F system was added to show that ZnO: ETP interlayer can improve the photovoltaic performance by extending the spectral response range of solar cells. In addition, we show the absorption spectral of PBDB-T-2F in Figure 10, which shows that PBDB-T-2F can absorb red light from re-emission of ZnO: ETP. We have added a discussion about PBDB-T-2F on page 13(line 361) of the manuscript.
Question 4: Concerning the degradation of polymer solar cells also the influence of morphological degradation has to be mentioned. This has been demonstrated in a number of studies including the PTB7-Th:PC71BM system with DIO as a solvent additive:
https://doi.org/10.1021/acsenergylett.8b02311
But again, the kind of device measured and presented in the plot of Figure 9 has to be specified.
Answer: Thank you very much for your good suggestions. In the description of polymer solar cell degradation[6] on page 1(line 39), we added a reference to the effect of morphological degradation and added a related citations.
Yang, D.; Löhrer, F.C.; Körstgens, V.; Schreiber, A.; Bernstorff, S.; Buriak, J.M.; Müller-Buschbaum, P. In-Operando Study of the Effects of Solvent Additives on the Stability of Organic Solar Cells Based on PTB7-Th:PC71BM. ACS Energy Lett. 2019, 4, 464-470.
Question 5: The caption of Figure 10 the meaning of arrows and red circles should be given. Although both systems are put into the figure, the discussion is reduced to the PTB7-Th system (line 337).
Answer: Thank you very much for your good suggestions. We have added a description of the red circles and arrows in Figure 10 and added a discussion about PBDB-T-2F on page 13(line 361) of the manuscript. The corrections in the manuscript are as follows:
In particularly, PTB7-Th has a strong absorption at 550-750 nm, and PBDB-T-2F has a strong absorption at 400-700nm.The absorption spectra of these two donor materials are match well with the red light (612 nm) re-emitted by the ETP complexes, so this red light can be absorbed by the active materials. Therefore, the ETP down-conversion material can effectively enhance the photoelectronic response of PSCs, extending the spectral response range of the PSCs to the UV region.
Figure 10. UV-visible absorption spectra of ETP, PTB7-Th:PC71BM and PBDB-T-2F:IT-4F(The red circle symbolizes the peak position in the ETP emission spectrum; the arrow symbolizes that UV light and visible light are absorbed by the ETP and active layer materials, respectively).
In particularly, PTB7-Th has a strong absorption at 550-750 nm, and PM6 has a strong absorption at 400-700nm.The absorption spectra of these two donor materials are match well with the red light (612 nm) re-emitted by the ETP complexes, so this red light can be absorbed by the active materials. Therefore, the ETP down-conversion material can effectively enhance the photoelectronic response of PSCs, extending the spectral response range of the PSCs to the UV region.

Reviewer 3 Report
The manuscript is ordered and clearly written. I believe this paper fits to the scope of the journal and will appeal to a broad range audience. Therefore, I can recommend the paper to be published, after one minor ambiguity is clarified: The long-term stability should be compared within two different condition: under the UV light and simulated light without UV (by using UV-cutoff lens) to investigate the effect of incorporated down-conversion material (ETP) on the durability of OPVs.
Author Response
The manuscript is ordered and clearly written. I believe this paper fits to the scope of the journal and will appeal to a broad range audience. Therefore, I can recommend the paper to be published, after one minor ambiguity is clarified: The long-term stability should be compared within two different condition: under the UV light and simulated light without UV (by using UV-cutoff lens) to investigate the effect of incorporated down-conversion material (ETP) on the durability of OPVs.
Answer: Thank you very much for this good suggestion. We measured the stability of the devices in simulated light without UV light and stored them in a dark room. Due to time, we measured 4 days of device efficiency decay data. Without UV light, the devices present more slower normalized efficiency reduction. In addition, the decay rate of ZnO device and ZnO: ETP device is basically the same.

Reviewer 4 Report
The manuscript entitled "Hybrid ZnO Electron Transport Layer by Down Conversion Complexes for Dual Improvements of Photovoltaic and Stable Performances in Polymer Solar Cells” focuses on the effect of the introduction of Eu(TTA)3phen (ETP) into the cathode transport layer (ZnO). The research undertaken by the Authors is important and interesting. The manuscript is well written. It can be accepted after improvement. The issues are listed below:
- Figure 1b: The structure of PTB7-Th is incorrect
- Figure 1c: The PL spectrum of ZnO:ETP is shown. It should be compared with the PL spectrum of ZnO. It is important for confirmation that ETP is the source of emission.
- Figure 2: The description of the axis should be made in larger fonts, similar to fonts on other figures.
- Line 159: It is not clear how the HOMO/LUMO values of ZnO:ETP (-7.63/-4.31 eV) were estimated, maybe it is from some references? It should be clarified.
Author Response
The manuscript entitled "Hybrid ZnO Electron Transport Layer by Down Conversion Complexes for Dual Improvements of Photovoltaic and Stable Performances in Polymer Solar Cells” focuses on the effect of the introduction of Eu(TTA)3phen (ETP) into the cathode transport layer (ZnO). The research undertaken by the Authors is important and interesting. The manuscript is well written. It can be accepted after improvement. The issues are listed below:
Question 1: Figure 1b: The structure of PTB7-Th is incorrect.
Answer: Thank you very much for the kind suggestions. We are very sorry for this mistake, after comparing with the structure drawing provided by Solarmer Materials Inc., we have corrected the structure of PTB7-Th in Figure 1 (b) in Page 4 as follow:
Figure 1. (a) Schematic illustration of the device architecture in this work. (b) The molecular structure of polymer PTB7-Th and PC71BM. (c) Absorption spectra of the neat PTB7-Th and PC71BM films and PL spectra of ZnO and ZnO:ETP. (d) Energy level diagram of the components of the device.
Question 2: Figure 1c: The PL spectrum of ZnO:ETP is shown. It should be compared with the PL spectrum of ZnO. It is important for confirmation that ETP is the source of emission.
Answer: Thank you very much for this good suggestion. We added the PL spectrum of ZnO in Figure 1 (c), we added some clarification to confirm that ETP is the source of emissions in Page 4(line 160). The description is as follow in the revised manuscript:
The absorption spectra of the neat PTB7-Th and PC71BM films and PL spectra of ZnO and ZnO:ETP are shown in Figure 1 (c), ZnO has weak fluorescence at 420nm-675nm, and there is no obvious peak at 612nm, and the ETP complexes are excited by UV light and re-emits red light (612 nm), indicating that ETP is the source of emission. In addition, it is matched well with the absorption of the donor material (PTB7-Th), thus helping to enhance photovoltaic performances in PSCs.
Question 3: Figure 2: The description of the axis should be made in larger fonts, similar to fonts on other figures.
Answer: Thank you very much for your suggestion. We have adjusted the font of the description of the axis in Figure 2 by comparing it with other figures to be consistent with the font in other figures in Page 5 as follow:
Question 4: Line 159: It is not clear how the HOMO/LUMO values of ZnO:ETP (-7.63/-4.31 eV) were estimated, maybe it is from some references? It should be clarified.
Answer: Thank you very much for the kind suggestions. We are very sorry for this mistake.The specific HOMO and LUMO values are uncertain, but considering that the amount of doping is extremely small, the energy level will not change too much. The energy level of ZnO:ETP in Figure 1 (d) is derived from the energy level of ZnO in the literature, which has been deleted in Figure 1 (d).

Round 2
Reviewer 1 Report
Question 1 did not explain the cause of the decrease in Voc.
Specifically, it is wrong to say that the Voc decreases due to the interface problem between Inorganic and organic. In many papers, the Voc of ZnO and PTB7-Th: PCBM is 0.80~0.81V. (DOI: 10.1039/C8RA08919A)
The explanation in question 3 is not enough.